# Biomarkers for Monitoring Treatment Response of Omalizumab in Patients with Chronic Urticaria

**DOI:** 10.3390/ijms241411328

**Published:** 2023-07-11

**Authors:** Nadja Højgaard Pedersen, Jennifer Astrup Sørensen, Misbah Noshela Ghazanfar, Ditte Georgina Zhang, Christian Vestergaard, Simon Francis Thomsen

**Affiliations:** 1Department of Dermato-Venereology and Wound Healing Centre, Copenhagen University Hospital Bispebjerg, 2400 Copenhagen, Denmark; nadja.hoejgaard.pedersen.01@regionh.dk (N.H.P.); jennifer.astrup.soerensen@regionh.dk (J.A.S.); misbah.noshela.ghazanfar@regionh.dk (M.N.G.); dzha0006@regionh.dk (D.G.Z.); simon.francis.thomsen.02@regionh.dk (S.F.T.); 2Department of Dermatology and Venereology, Aarhus University Hospital, 8200 Aarhus, Denmark; 3Department of Biomedical Sciences, University of Copenhagen, 2200 Copenhagen, Denmark

**Keywords:** chronic urticaria, chronic spontaneous urticaria, biomarkers, omalizumab, anti-IgE, treatment response

## Abstract

Chronic urticaria (CU) is a debilitating skin disease affecting around 1% of the population. CU can be subdivided into chronic spontaneous urticaria (CSU) and chronic inducible urticaria (CIndU). Different pathophysiological mechanisms have been proposed to play a role in the development of CU, and these are also being investigated as potential biomarkers in the diagnosis and management of the disease. As of now the only assessment tools available for treatment response are patient reported outcomes (PROs). Although these tools are both validated and widely used, they leave a desire for more objective measurements. A biomarker is a broad subcategory of observations that can be used as an accurate, reproducible, and objective indicator of clinically relevant outcomes. This could be normal biological or pathogenic processes, or a response to an intervention or exposure, e.g., treatment response. Herein we provide an overview of biomarkers for CU, with a focus on prognostic biomarkers for treatment response to omalizumab, thereby potentially aiding physicians in personalizing treatments.

## 1. Introduction

Chronic spontaneous urticaria (CSU) is a skin disease characterized by the formation of wheals and/or angioedema for 6 weeks or more accompanied by itching, burning, and sometimes painful sensations in the skin. Wheals are edema in the upper part of the dermis, typically surrounded by reflex erythema and characterized by a fleeting nature, lasting only for a few hours (<24 h). Angioedema is edema located in the lower part of the dermis and in the subcutis, often remaining there for more than 24 h. CSU can cause severe distress in patients due to bothersome symptoms and the unpredictable nature thereof, potentially leading to anxiety in addition to other comorbidities [1,2].

CSU is caused by mast cell activation and degranulation with the release of histamine as the main mediator of symptoms alongside other cytokines and neuropeptides. The release of histamine from mast cells is a classic type I immediate reaction in the Coombs and Gell’s classification, caused by the cross binding of two immunoglobulin(Ig)E molecules by an allergen, which in turn are bound to the Fc-receptor on the surface of the mast cells.

The first-line therapy of CU is second generation non-sedating H1-antihistamines, which may be used in up to quadruple dosage according to the EAACI/GA2LEN/WAO guidelines. Omalizumab is introduced into the treatment if there is an inadequate response to first-line therapy [1]. Omalizumab is a monoclonal anti-IgE antibody that prevents IgE from binding to the FcεR1 receptor on mast cells and basophils, thereby inhibiting the release of inflammatory mediators [3]. However, with new emerging drugs for the treatment of CU it has become clear that the IgE-associated pathway is not the only pathway involved in the pathogenesis of CU.

In CSU the type I reaction or auto allergic CSU (aaCSU) also accounts for the majority of cases, but in contrast to classic allergies the allergen is an autoallergen, e.g., thyroid peroxidase (TPO) or interleukin (IL)-24. A less common cause of mast cell activation is the cross binding of the Fc-receptors or the IgE molecules by IgG or IgM auto-antibodies, known as a type IIb reaction or autoimmune CSU (aiCSU) [1]. Intracellularly the signal from the cross binding is mediated through the Bruton tyrosine kinase (BTK) pathway that activates nuclear factor kappa-light-chain-enhancer (NF-κB), nuclear factor of activated T cells (NFAT), and activator protein-1 (AP-1). The BTK is a target for emerging kinase inhibitors being tested in CU [4]. An emerging antibody for the treatment of CU, barzolvolimab, targets the KIT receptor on the surface of mast cells and blocks the binding of stem cell factor (SCF) which is needed for the activation of mast cells [5]. Another new antibody, lirentelimab, targets Sialic Acid Binding Immunoglobulin-like Lectin 8 (Siglic 8) and inhibits the activity of the mast cells and thereby also CU. Finally, the anti-IL-4Ra antibody dupilumab has also shown efficacy in the treatment of CSU. Taken together, all these new results demonstrate the complexity of the pathogenesis of CU.

New therapies offer the possibility of more efficient treatment, but also personalized medicine where one size does not fit all and where treatment is based on the deep phenotyping of the patients. To achieve this, biomarkers are needed. Biomarkers are by definition very diverse and may be clinical signs and symptoms, or they may be measurable markers either physical, such as an ECG, or biochemical, such as cholesterol [6,7]. Based on their function there are several different types of biomarkers. Biomarkers may be used for assessing risk, aiding in diagnosis, monitoring disease activity or treatment efficiency, predicting treatment response, disease duration, and severity, or evaluating the safety of a given drug.

As our understanding of the pathogenesis of CU is expanding and our options for treatment via different pathways are growing, we need to explore the field of biomarkers for CU, especially prognostic biomarkers and biomarkers for the treatment response of omalizumab, to perform personalized medicine. Herein we provide an overview of the relevant literature for selected laboratory biomarkers with a focus on treatment response in CU but also touching upon the subject of disease activity.


**Methods**


Biomarker selection: The biomarkers in the basic diagnostic workup group were chosen based on the current international recommendations on diagnostic workup in CU [1]. The extended workup and emerging biomarkers groups were chosen by the authors based on their knowledge in the field.

This review is not systematic but an evaluation of relevant and carefully selected literature on the chosen biomarkers.

## 2. Biomarkers in the Basic Diagnostic Workup of CU

According to the current guidelines [1] for CU, there are several biomarkers, which are useful in diagnostic workup and are predictive for treatment response in CU. In the following, some of the relevant biomarkers related to basic diagnostic workup and omalizumab response in patients with CU will be presented. See Table in Section 4.5 for an overview.

### 2.1. IgG Anti-Thyroid Peroxidase (IgG Anti-TPO)

IgG anti-TPO antibodies are autoantibodies that target the thyroid peroxidase enzyme, which plays a crucial role in the production of thyroid hormones. The presence of TPO-antibodies is typically associated with thyroid disorders such as Hashimoto’s disease [8].

The relationship between IgG anti-TPO and CU has been investigated in some studies which suggest that the presence of IgG anti-TPO antibodies in patients with CU is indicative of aiCSU. The aiCSU subtype is characterized by high disease activity, the presence of concomitant comorbidities, poor quality of life, and poor response to treatment with non-sedating antihistamine and omalizumab [9]. In a study by Schoepke et al. on biomarkers for aiCSU it was reported that IgG anti-TPO levels were significantly higher and that the rates of elevated IgG anti-TPO were greater in patients with aiCSU [10].

In a retrospective study of 138 CU patients the proportion of patients with elevated IgG anti-TPO levels was higher in non-responders to omalizumab compared to responders (44.4% vs. 14.9%). The study suggested that having elevated IgG anti-TPO may be associated with being a non-responder to omalizumab [11].

In another study, IgG anti-TPO was investigated as a potential biomarker for response to omalizumab treatment. The study comprised 385 patients (262 females) with severe CSU. Total IgE levels and thyroid autoimmunity were measured before treatment with omalizumab. A total of 92 patients (24%) had thyroid autoimmunity. The study concluded that response to omalizumab was not associated with thyroid autoimmunity and that thyroid autoimmunity on its own could not be used as a predictive marker for poor response to omalizumab treatment [12].

Taken together these studies are inconclusive for IgG anti TPO as a biomarker for response to omalizumab therapy.

### 2.2. Total Immunoglobulin E (IgE)

IgE is an antibody which plays a significant role in the immune system’s response to allergens and parasites. Omalizumab reduces the free level of IgE, leading to the downregulation of IgE receptors on mast cells and basophils in CU. On average, IgE levels are increased in patients with aaCSU (type I) but lowered in patients with aiCSU (type IIb). Total serum (s-) IgE levels can therefore be considered a predictor for response to omalizumab treatment in patients with CU [13].

Several studies have investigated total s-IgE levels and response to omalizumab in patients with CU [13,14,15,16]. Total s-IgE levels were investigated before and after treatment in a German study with 133 antihistamine refractory CSU patients, who were treated with the standard dose of omalizumab every fourth week for 12 weeks. Response to treatment was evaluated with the urticaria activity score (UAS7). After 12 weeks of treatment, it was seen that non-responders had lower total s-IgE levels at baseline as well as a lack of increase in total s-IgE levels when compared to partial responders and complete responders [16]. Similar observations were made in a study by Cugno et al. with 25 CSU patients treated with omalizumab [14]. Furthermore, it was seen that total s-IgE levels increased significantly in partial and complete responders within 4 weeks of omalizumab treatment due to the formation of IgE–omalizumab complexes [16]. In a prospective study, Ertas et al. investigated whether elevated levels of total s-IgE in CSU patients could be related to relapse of symptoms. It was seen that elevated levels of total s-IgE seem to be a predictor for a faster relapse of urticaria symptoms if treatment is terminated [17]. On the other hand, low total s-IgE levels at baseline and higher disease activity were seen more commonly in non-responders [15]. A study by Oda including 34 CSU patients observed that total s-IgE tended to be higher in fast responders but with no statistically significant difference [18]. Another study by Ghazanfar et al. including 120 patients with CSU did not find a statistically significant association between total s-IgE levels and response to omalizumab [19]. Pinto Gouveia et al. also found no laboratory factors predicting response to omalizumab treatment [20].

### 2.3. C-Reactive Protein

C-reactive protein (CRP) is an acute-phase reactant which is elevated in the presence of inflammation. CRP is not considered a primary biomarker related to omalizumab treatment response; however, recent studies have suggested that CRP may be increased in CU due to mast cell activation, which explains why elevated levels are commonly found in patients with CU [21]. Elevated levels of CRP in patients with CU have previously been linked to non-response to antihistamines [22].

In a recent study [23] the long-term effect of omalizumab on peripheral blood and serum CRP was investigated. The study included 74 CSU patients treated with omalizumab continuously for 1 year. A significant reduction in CRP levels was seen after three months of treatment with omalizumab. Additionally, it was seen that CRP levels remained lower than baseline levels at 12 months follow-up. In another study, it was seen that CRP levels significantly correlated with disease activity (UAS7) at baseline and after treatment with omalizumab, indicating anti-inflammatory effects of omalizumab [24].

### 2.4. White Blood Cell Count and Platelets

Basophils have been linked to the pathogenesis of CSU, with basophil numbers and urticaria severity inversely related [25]. In a double-blinded study by Metz et al. [26], the effect of omalizumab was investigated on IgE and basophils. Mean levels of basophils were lower in untreated patients with CSU, and an increase in basophil count was seen after treatment with omalizumab compared to the placebo group. In another recent study, a significant increase in basophils was seen after treatment with omalizumab at 12 months follow-up compared to baseline [23].

Eosinophils are increased in allergic responses and are involved in the pathogenesis of asthma and other allergic diseases. Eosinophils are increased in CSU and found in both lesional and non-lesional skin of patients with CSU. Furthermore, it was seen in a study that low blood eosinophil counts were associated with non-response to omalizumab, although it is important to take other diseases (e.g., malignancy) causing eosinopenia into account [27].

Recently the use of inflammatory biomarkers based on white blood cell count and platelets has gained increased interest due to its cost-effectiveness and accessibility. These biomarkers include the systemic inflammatory response index (SIRI), the systemic immune-inflammation index (SII), the platelet lymphocyte ratio (PLR), and the neutrophil/lymphocyte ratio (NLR), which are calculated based on levels of neutrophils, lymphocytes, monocytes, or platelets. Some studies have demonstrated a significant reduction in these biomarkers following omalizumab treatment [28,29,30]. A study conducted by Consansu et al., including 124 patients with CSU, proposed SII and SIRI as reliable predictors of response to omalizumab. The study found that responders exhibited significantly higher SII and SIRI levels compared to non-responders at three and six months after initiating omalizumab treatment [29]. However, a recent study failed to verify the utility of these biomarkers [31].

## 3. Biomarkers in the Extended Workup of CU

In addition to the biomarkers recommended in guidelines for initial diagnostic evaluation, other biomarkers have also been identified as potentially helpful in predicting treatment outcomes. We present some of these here, as they may support the basic diagnostic workup and are accessible in routine clinical practice. See Table in Section 4.5 for an overview.

### 3.1. IgG Anti-Thyroglobulin (IgG Anti-TG)

Antibodies to thyroglobulin (IgG anti-TG) are also markers of Hashimoto’s disease [32], alongside IgG anti-TPO, although with a lower sensitivity and specificity [33]. IgG anti-TG is, however, still a relevant biomarker to assess as a meta-analysis by Pan et al. estimated the odds ratio (OR) for IgG anti-TG for patients with CSU to be 6.55 (95% CI: 3.19–13.42) compared to healthy controls [34]. The study also revealed that this marker was primarily associated with urticaria within the Asian population, OR 8.02 (95% CI 3.58–17.97), and not the European population, OR 3.64 (95% CI 0.67–19.83). A Turkish study by Cakmak et al. with 220 CU patients (154 women) found that after six months of treatment with omalizumab non-responders were significantly more likely to have positive anti-TG at baseline compared to responders (35.5% vs 13.2%) [35]. Although the prevalence of autoimmune comorbidities in this cohort was rather high (20.5%), it may suggest TG-Ab as a potential biomarker for omalizumab response, but further studies are needed to confirm the value of TG-Ab as a predictor for treatment response.

### 3.2. BHRA

Basophil histamine release assay (BHRA), also called HR urticaria test, is currently the gold standard for identifying aiCSU [36]. The test measures the histamine release of the total overall histamine content, where a value >16.5% indicates a positive test for both adults and children [37].

In relation to omalizumab treatment response a study by Gericke et al. including 64 patients (46 women) found that BHRA positive patients were more likely to be late responders to omalizumab compared to BHRA negative patients, with a median time of 29 days vs. 2 days [38]. The poor treatment response associated with a positive BHRA is supported by a study by Ghazanfar et al. which included 154 patients (110 women). The study found that a negative BHRA was significantly associated with complete or almost complete response to omalizumab after 3–6 months’ treatment, OR 9.07 (95% CI 2.80–29.32), compared to BHRA positive CSU patients [39].

### 3.3. D-Dimer

D-dimer is a product of cross-linked fibrin degradation and is considered a biomarker for coagulation activation and fibrinolysis, playing an important role when ruling out venous thromboembolism. Because D-dimer is a reliable biomarker for fibrin digestion and coagulation activation, D-dimer testing is used in several clinical conditions, making it a less specific biomarker [40]. Significant changes in coagulation biomarkers such as D-dimer have been observed in patients with CU during exacerbations and after remission, making it a possible biomarker for treatment response [41].

A study by Asero et al. assessed D-dimer levels in 32 adults with severe CSU before and after the first administration of omalizumab. The study showed that in all responders D-dimer levels decreased after the first administration of omalizumab (from 1024 ± 248 [mean ± SE] to 251 ± 30 ng/mL). In non-responders an insignificant increase in D-dimer levels (787 ± 206 to 1230 ± 467 ng/mL) was observed [42]. Other studies have found similar correlations [22,43,44]. A study by Marzano et al. including 470 CSU patients found that baseline D-dimer was not associated with clinical response to omalizumab as the mean values were similar in responders and non-responders [13].

### 3.4. Antinuclear Antibodies

Antinuclear antibodies (ANA) are antibodies that react to components within the cell nucleus. They are commonly used to detect autoimmune connective tissue disease [45]. In patients with CU, ANA is primarily used to screen for autoimmune comorbidities. Although several studies have reported a higher prevalence of ANA positivity among patients with CU, a review by Kolkhir et al. revealed that the prevalence varies widely, ranging between 0–31.9%, and that most studies consisted of small sample sizes [46], thus making the clinical implications of ANA positivity unclear.

To investigate the relevance of ANA positivity in patients with CU in relation to omalizumab treatment, Ertaş et al. conducted a study on 171 patients with CSU. The study found that ANA positive patients experienced a lower mean reduction in disease activity (assessed by visual analogue scale, VAS) 12 weeks after treatment initiation compared to ANA-negative patients (33% vs. 95%). Furthermore, there was a significant negative correlation between ANA titers and treatment response to omalizumab (r = −0.345), and ANA positive patients were significantly more likely to be non-responders compared to ANA negative patients (45% vs. 9%) [47]. These results indicate that ANA may be utilized to support the presence of aiCSU and indicate a poorer response to omalizumab treatment in these patients, although further studies are needed to confirm these findings.

## 4. Emerging Biomarkers for Treatment Response

The research field of biomarkers in patients with CU is ever expanding due to the lack of precise and accurate biomarkers for diagnosis, prognosis, and treatment response. Here we present a short overview of emerging biomarkers and specialized biomarkers that may not be accessible or common in routine clinical practice. See Table 1 for an overview.

### 4.1. Adenosine

More recently adenosine, a modulator of mast cell function, has shown potential for assessing omalizumab effectiveness. A study by Mao et al. found significantly higher mean plasma levels of adenosine in patients with CSU when compared with controls and significantly higher levels in patients with severe disease when compared to mild and moderate disease [48]. Furthermore, patients with severe disease who were non-responders to antihistamines had higher plasma levels of adenosine than responders. These results suggest that adenosine may be a useful predictor of disease severity and possibly treatment response, although more studies are needed to further investigate these findings.

### 4.2. FcεRI Receptor

The FcεRI receptor, a high-affinity IgE receptor, is located on both basophils and mast cells and plays a key role in the pathogenesis of CU [49]. Studies have demonstrated that reduction in the receptor expression was significantly associated with clinical improvement after initiation of omalizumab treatment [26,50]. This suggests that monitoring changes in FcεRI receptor expression could serve as a potential biomarker for assessing treatment response. However, four studies all showed no relation between clinical response to omalizumab and reduction in FcεRI receptor expression, finding only a significant reduction in expression after treatment initiation that did not differ between the groups [51,52,53,54]. Another set of studies from the same prospective cohort found that CU patients had higher levels of FcεRI compared to healthy controls, but the subgroups of non-responders to omalizumab had lower baseline levels of FcεRI than the healthy controls. This could indicate that baseline levels of FcεRI could be a predictor for non-response to omalizumab [55,56,57,58]. A different study found no difference in baseline values between responders and non-responders to omalizumab [18].

### 4.3. Interleukins

Furthermore, IL-31, a cytokine commonly associated with pruritic skin diseases [59,60], has also been suggested as a biomarker for monitoring treatment response in CU, as a study found that s-IL-31 levels were significantly reduced after successful treatment with omalizumab [61]. One study found that patients with CU had higher levels of IL-31 than healthy controls [62] but lower values than patients with atopic dermatitis, whereas another study did not find elevated levels of IL-31 in CSU patients compared to controls [63].

Another interleukin recently investigated in CU is IL-6, which has been shown to play a key role in chronic inflammation [64]. A pilot study of eight patients with CSU receiving omalizumab and four controls found that the patients with CSU had elevated s-IL-6 compared to controls at baseline. After six months of treatment s-IL-6 levels were significantly lower compared to baseline values [65]. Another study of 58 patients with CU compared to 30 controls also found an increase in s-IL-6 in the patients with CU, with a decrease in levels after spontaneous remission [66]. This could indicate that s-IL-6 is more of a predictor of disease activity rather than treatment response. This was also a point in a study by Rasool et al., who found a difference in IL-6 levels between severity groups where patients with the most severe urticaria also had the highest levels of IL-6. However, they did not find a significant difference between patients with CU and controls and did not examine treatment response [67]. Metz et al. also found no difference between patients with CSU and controls with regards to IL-6 levels [63]. Therefore, further studies are needed to determine IL-6’s possible role in CU.

### 4.4. Transglutaminase 2

A recent study also proposed transglutaminase (TG) 2 activity as a potential biomarker. In a murine asthma model TG2 was involved in mast cell activation, IgE production, and the release of mediators [68]. Bae et al. showed a significantly increased level of TG2 activity in CSU patients compared to controls, and when looking at CSU severity they found increasing activity with increasing severity [69]. In a small subpopulation of nine omalizumab-treated patients they showed a significant decrease in TG 2 activity 2–6 weeks after treatment initiation and further decrease after 16–24 weeks, making it a potential biomarker for monitoring treatment response. In a very recent study IgE-anti-TG2 levels were elevated in 20% of CSU patients [70], thus pointing out the need for further studies into TG2 and its role in the pathogenesis of CU and potential clinical relevance.

### 4.5. Basophil CD203c

Basophil CD203c is marker for basophil activation that is upregulated by the cross-linking of the FcεRIα receptor [71]. Moreover, CD203c was found to be expressed selectively on the blood basophils and mast cell surfaces and their CD34 progenitor cells [72]. The ability to upregulate CD203c has been proposed as a predictor of treatment response as a study by Palacios et al. found that CSU patients without CD203c-upregulating activity were more likely to have a good treatment response [73]. Furthermore, these patients also responded faster compared to CSU patients without upregulated CD203c activity. When looking at CD203c responsiveness to stimulation through the FcεRI Oda et al. found that fast responders to omalizumab had an increased CD203c responsiveness after omalizumab treatment when compared to non- or slow responders [18] and there were no baseline differences between the two groups. Further investigation into CD203c is needed to determine whether it is a useful predictor of treatment response to omalizumab.

**Table 1 ijms-24-11328-t001:** Biomarkers associated with omalizumab response and disease activity in patients with chronic spontaneous urticaria.

Biomarker	Increase/Decrease	Meaning	Studies Are	Reference	CU Patients Investigated
**IGG-ANTI-THYROID PEROXIDASE (ANTI-TPO)**	Increase	Associated with aiCSU and non-response to omalizumab	Contradictory	[10]	182
				[11]	138
				[12]	385
**SERUM IGE LEVELS**	Decrease	Predictive of response to omalizumab	Consistent	[13]	470
				[14]	25
				[15]	137
				[16]	113
				[17]	93
				[18]	34
				[19]	120
				[20]	13
**C-REACTIVE PROTEIN (CRP)**	Increase/decrease	Associated to disease activity	Consistent	[23]	74
				[24]	147
**BASOPHIL COUNT**	Increase/decrease	Associated to disease activity	Consistent	[23]	74
				[26]	20 *
**EOSINOPHIL COUNT**	Decrease	Predictive of non-response to omalizumab	One study	[27]	1613
**SIRI/SII/PLR/NLR**	Increase	Predictive of response to omalizumab	Contradictory	[28]	252 (106 completed)
				[29]	124
				[30]	143
				[31]	56
**IGG-ANTI-TG**	Increase	Predictive of non-response to omalizumab	Consistent	[34]	Review 13.898 ^†^
				[35]	220
**BASOPHIL HISTAMINE RELEASE ASSAY (BHRA)**	Positivity	Predictive of non-response to omalizumab	Consistent	[38]	64
				[39]	154
**D-DIMER**	Decrease	Predictive of response to omalizumab	Inconsistent	[13]	470
				[22]	95
				[42]	32
				[43]	8 ^†^
**ANTI NUCLEAR ANTIBODIES**	Positivity	Predictive of non-response to omalizumab	One study	[47]	447 (171 treated with oma)
**ADENOSINE**	Increase/decrease	Associated to disease activity	One study	[48]	48 ^†^
**FCΕRI RECEPTOR EXPRESSION ON BASOPHILS**	Increase	Predictive of response to omalizumab	Inconsistent	[18]	34
				[26]	18
				[50]	30 (20 oma) *
				[51]	30 (25 completed) *
				[52]	30 CSU
				[53]	15
				[54]	18
				[55]	47 ^†^
				[56]	44
				[57]	287 ^†^
				[58]	165 ^†^
**INTERLEUKIN 31**	Increase	Associated to disease activity	Inconsistent	[61]	39 (21 oma) *
				[62]	46 ^†^
				[63]	58 ^†^
**INTERLEUKIN 6**	Increase	Associated to disease activity	Inconsistent	[63]	58 ^†^
				[65]	8 ^†^
				[66]	58 ^†^
				[67]	62 ^†^
**TRANSGLUTAMINASE 2**	Decrease	Associated to disease activity	Consistent	[69]	111 (31 AU) ^†^
				[70]	160 ^†^
**BASOPHIL ACTIVATION TEST (CD203C EXPRESSION)**	Increase	Predictive of response to omalizumab	Consistent	[18]	34
				[73]	41

* Placebo controlled study; ^†^ Controls.

## 5. Discussion

Overall, the studies of biomarkers reviewed herein are of varying size and study design. Therefore, a general theme is that further studies are needed to establish the utility of the biomarkers. In the following we present a short discussion on the diagnostic tests presented above.

### 5.1. Basic Diagnostic Workup

IgG anti-TPO: two studies were presented with conflicting results with regards to IgG anti-TPO as a biomarker for response to omalizumab therapy [11,12].

Serum total IgE: quite a few studies have investigated the link between IgE and response to omalizumab. Here we presented eight studies of which the majority found a link between IgE and omalizumab response [13,14,15,16,17,18,19,20]. Even though few studies found no association, two of these had a low number of participants and may lack the power to accurately determine an association. All in all, the studies finding a connection between IgE and omalizumab response were larger cohorts of patients; therefore, based on the presented studies it is evident that lowered levels of total s-IgE before initiating omalizumab treatment can be used as a predictor for poor or non-response to omalizumab treatment.

CRP: While CRP correlates with disease activity [23,24], it is not a useful predictor for response to omalizumab treatment. However, it has been suggested that CRP is more useful in differential diagnosis, where a high CRP level that does not decrease during treatment could be an indicator of auto-inflammatory syndrome [22].

White blood cell count and platelets: Only very few studies investigated these as biomarkers for omalizumab response. The two studies on basophils both showed an increase in basophils after omalizumab treatment, and one was a placebo-controlled trial [23,26]. All in all, the studies on SIRI, SII, PLR, and NLR are conflicting [28,29,30,31].

All the tests above are routine tests conducted in most laboratories all over the world, and their price point is at the lower end of the scale. They are also recommended in the basic diagnostic workup of CSU. Therefore, even though there may be only few studies and conflicting results on these tests as biomarkers for treatment response to omalizumab they still play a vital role in urticaria diagnosing and are relatively cheap. Physicians do have to bear in mind that these biomarkers are not specific for urticaria and are used widely in diagnostic workup, for example, in infections or for basophils, where basopenia can also indicate malignancy.

### 5.2. Extended Diagnostic Work Up

IgG anti-TG: A meta-analysis found that CSU patients had higher odds of having IgG anti-TG compared to healthy individuals, but it did not examine treatment response [34]. Only one study looked at treatment response, and it indicated that non-responders to omalizumab treatment were more likely to have positive anti-TG at baseline [35]. There was, however, a relatively high prevalence of autoimmune comorbidities in this cohort.

BHRA: Two studies examined treatment response to omalizumab in relation to BHRA; they found that late or poor response were associated with a positive BHRA [38,39]. Looking at these two studies, BHRA provides valuable insight into the patient’s likely response to omalizumab treatment, in terms of both the likelihood and the timing of their response. Thus, BHRA may assist physicians in personalizing aiCSU treatment plans. That said, BHRA is a specialized test not readily available as of now, making its potential as a biomarker limited.

D-dimer: Two smaller and one slightly larger study found a decrease in D-dimer levels in omalizumab responders [22,42,43]. One larger study found no association between baseline D-dimer levels and omalizumab response [13]. It is important to note that D-dimer was analyzed at different timepoints in the studies. It is also imperative to highlight the minimal specificity of this biomarker and its use in several clinical conditions.

ANA: Various studies have reported a higher prevalence of ANA positivity in CU patients, but the prevalence varies extensively, and a significant limitation is the small sample sizes of the studies [46]. Only one study examined treatment response to omalizumab and ANA status where ANA positive patients were more likely to be non-responders [47]. This indicates that ANA may be used as a potential biomarker to support the diagnosis of autoimmune CU (aiCSU) and predict a poorer response to omalizumab treatment in these patients. However, these results should be taken cautiously as further studies with larger sample sizes are needed to validate these findings and explore more in-depth the relationship between ANA positivity, disease activity, and response to omalizumab treatment.

The tests presented above are not as routine as in the basic workup but are quite widely used in other diseases. They are also around the same price point but a bit more expensive than the basic workup. An exception is BHRA, which is not widely available and is more expensive, but also more specific to urticaria, than the other biomarkers in this category. Before acquiring use as clinical tests, specific ranges for these biomarkers need to be established in order for threshold ranges for clinically relevant tests to be established.

### 5.3. Emerging Biomarkers

Adenosine: this biomarker has only been investigated in one study and only in regard to disease severity and antihistamine response. These results suggest that adenosine may be a useful predictor of disease severity and possibly treatment response.

FcεRI: Eleven studies were found to have investigated FcεRI expression in relation to omalizumab response [18,26,50,51,52,53,54,55,56,57,58]. Some look at expression before and after treatment and others at baseline expression. The four articles showing that CU patients had higher baseline levels compared to healthy controls except for the sub-group of non-responders who had lower baseline levels compared to the control group seem to be from the same cohort of patients [55,56,57,58]. Therefore, it could be argued that these should be considered as one study rather than four. All the other studies on the FcεRI receptor investigated a relatively small number of patients, and overall the results are conflicting. The two studies showing a significant association between clinical improvement after omalizumab treatment and reduction in receptor expression were both randomized controlled trials, increasing their level of evidence.

IL-31: Only three studies investigated the relation between IL-31 and CU [61,62,63]. The two studies that looked at levels of IL-31 in both patients with CU and controls were of similar size with regards to number of patients with CU but differed in number of controls, and the results were conflicting. Only one study looked at treatment response, and it found lower levels after successful treatment; therefore, IL-31 might be more of a biomarker for disease activity, but the number of studies is limited.

IL-6: Two of the four studies on IL-6 found higher baseline values in patients with CU compared to controls, and the other two did not find a significant difference [63,65,66,67]. When looking at treatment response the two studies with higher baseline levels found a significant decrease in levels after treatment. The other two studies did not look at omalizumab treatment. One of the studies that did not find a difference in baseline values did find higher levels of IL-6 in more severe cases of urticaria, but it is essential to note that all of the studies had a relatively small number of patients included.

Transglutaminase 2: Only two studies on this biomarker were found [69,70]. One found that TG2 levels were elevated in CSU patients compared to controls. The other study looked at IgE anti-TG2, which was elevated in one in five patients with CSU.

Basophil CD203c: One study looking at CD203c upregulating activity and one looking at CD203c responsiveness both found that there was a difference in these parameters when looking at responsiveness to omalizumab treatment [18,73]. Further investigation into CD203c is needed to determine whether it is a useful predictor of treatment response to omalizumab as the studies presented looked at different entities and did not include a large number of patients or any controls.

With regards to the emerging biomarkers and their utility as biomarkers for omalizumab treatment response further studies are needed. As of now most of the biomarkers are not widely available and are more expensive than the other two groups. Most of the biomarkers have only been investigated in very few studies, except for the FcεRI receptor, which has been looked at more extensively, yet even though eleven studies investigated this the evidence is still inconsistent. Therefore, it will probably be some time before we see these biomarkers in the basic diagnostic work-up for CU, if ever.

### 5.4. Limitations

The studies identified in this review were very heterogeneous, of varying quality and methodology, and usually small in size, with no control group; therefore, the results presented are not further analyzed. Another limitation is that this is not a systematic review, and therefore some relevant articles may have been missed.

## 6. Conclusions

To summarize, various biomarkers can be used to predict omalizumab treatment response.

Biomarkers associated with aiCSU indicate a slower and poorer response to omalizumab treatment, but as of now there is no single specific biomarker that can accurately predict treatment outcomes of omalizumab in patients with CU. Further research is therefore needed to identify more sensitive biomarkers.

## Data Availability

Not applicable.

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
