# Peer review of "Biomarkers for Monitoring Treatment Response of Omalizumab in Patients with Chronic Urticaria"

_ijms, 2023, doi:10.3390/ijms241411328_

Round 1
Reviewer 1 Report
The authors present a comprehensive review on biomarkers for monitoring of disease activity and treatment response to omalizumab in patients with CU. The subject is relevant and of high interested. Although reviews on CU biomarkers are published before an update and good overview is very useful.
However, given the huge amount of literature on CU biomarkers the authors do not explain the methodology followed, why specific biomarkers where chosen to review and which strategy is used for selection of articles. Subsequently, the quality of this comprehensive review remains uncertain.
E.g. the authors state that baseline level of FcεRI expression on basophiles could predict omalizumab response, based on the presented articels, which seems not correct for me. 3 out 6 cited articles come from the same group, while (at least) two relevant studies (Johal et al. and Oda et al), that did not found a relation between baseline basophil FcεRI expression and clinical omalizumab response are missing (PMID: 33713769; PMID: 32898709) . Furthermore, reference 46 (Alizadeh et al) seems interpreted incorrectly , since this study shows no relation between baseline basophil FcεRI expression or reduction during treatment and clinical omalizumab response instead of confirming this relationship.
Also studies investigating total IgE as biomarker, but do not prove relationship with clinical omalizumab response are not mentioned (PMID: 32898709 and others).
In general, quality of the described studies or differences in methodology, well or not explaining differences in outcome, is not described, compared or critically discussed.
For some specific biomarkers as total IgE , D-Dimer, WBC, CRP discussion about threshold levels could be useful, making it more practical.
A general discussion with final reviewing the findings, relating to clinical practice , feasibility and cost-effectiveness would be useful.
Together, the manuscript looks rather superficial and careful revision is recommended.
Specific comments:
- Throughout the manuscript phrasing as ”some studies demonstrated ..” should be more concrete .
- Legend for the table might be useful
- Abstract and introduction explain aiming to present prognostic biomarker predicting treatment response, while also disease activity seems one of the major objectives.
- line 36 : anti- H2 antihistamines are stated as first line treatment, is H1 antihistamine meant?
good
Author Response
We have inserted the response to both reviewers in the table in the attached word document

Reviewer 2 Report
The authors of the paper presented an overview of potential biomarkers used in chronic spontaneous urticaria, divided into laboratory biomarkers used in the basic and extended diagnostic work-up and newly emerging biomarkers.
The work is a valuable review of current publications on this topic. However, there are a few points that I think need to be clarified or improved:
= The authors limited themselves to laboratory biomarkers, although as they write - any observation related to the studied phenomenon (symptom, ECG result, laboratory result) can be a biomarker. If we limit ourselves to the results of laboratory measurements - let's make it clear in the text.
= What papers were selected for this review? - subjective choice of authors or defined search criteria in a defined database (like PubMed), which would enable further updating in the future and comparison with other reviews in the field
= No table title - what does it represent …
= Studies on the topic are relatively few and of varying quality in terms of the level of evidence (from small groups to large cohorts of patients). If authors do not further analyze their quality, they should mark this as a limitation of their review.
= So-called biomarkers such as WBC or DDimers have practically minimal specificity (increased value in many health situations not related to urticaria - from a common cold to pulmonary embolism and venous thromboembolism) this should absolutely be emphasized.
Author Response
We have combined our responses in one table
Round 2
Reviewer 1 Report
General comment:
The manuscript is substantially improved and describes finding from literature adequately. A discussion is added with nuancing the findings in perspective of the daily practice situation.
Minor specific comments :
Line 206 is not readable
no comment